# Association of Beverage Consumption during Pregnancy with Adverse Maternal and Offspring Outcomes

**DOI:** 10.3390/nu16152412

**Published:** 2024-07-25

**Authors:** Zhengyuan Wang, Xin Cui, Huiting Yu, Ee-Mien Chan, Zehuan Shi, Shuwen Shi, Liping Shen, Zhuo Sun, Qi Song, Wei Lu, Wenqing Ma, Shupeng Mai, Jiajie Zang

**Affiliations:** 1Department of Nutrition and Health, Division of Health Risk Factors Monitoring and Control, Shanghai Municipal Center for Disease Control and Prevention, Shanghai 200336, China; wangzhengyuan@scdc.sh.cn (Z.W.); shizehuan@scdc.sh.cn (Z.S.); shenliping@scdc.sh.cn (L.S.); sunzhuo@scdc.sh.cn (Z.S.); luwei@scdc.sh.cn (W.L.); mawenqing@scdc.sh.cn (W.M.); maishupeng@scdc.sh.cn (S.M.); 2Shanghai Health Statistics Center, Shanghai 200040, China; monicasnail@163.com; 3Division of Vital Statistics, Institute of Health Information, Shanghai Municipal Center for Disease Control and Prevention, Shanghai 200336, China; huitingyu@scdc.sh.cn; 4School of Public Health, Shanghai University of Traditional Chinese Medicine, Shanghai 201203, China; 18616535191@163.com; 5The College of Medical Technology, Shanghai University of Medicine and Health Sciences, Shanghai 200237, China; 15950475263@163.com

**Keywords:** sugar-sweetened beverages, non-sugar sweetened beverages, pregnancy, macrosomia, large for gestational age, offspring

## Abstract

Background: As the global consumption of sugary and non-sugar sweetened beverages continues to rise, there is growing concern about their health impacts, particularly among pregnant women and their offspring. Objective: This study aimed to investigate the consumption patterns of various beverages among pregnant women in Shanghai and their potential health impacts on both mothers and offspring. Method: We applied a multi-stage random sampling method to select participants from 16 districts in Shanghai. Each district was categorised into five zones. Two towns were randomly selected from each zone, and from each town, 30 pregnant women were randomly selected. Data were collected through face-to-face questionnaires. Follow-up data on births within a year after the survey were also obtained. Result: The consumption rates of total beverages (TB), sugar-sweetened beverages (SSB), and non-sugar sweetened beverages (NSS) were 73.2%, 72.8%, and 13.5%, respectively. Logistic regression analysis showed that compared to non-consumers, pregnant women consuming TB three times or less per week had a 38.4% increased risk of gestational diabetes mellitus (GDM) (OR = 1.384; 95% CI: 1.129–1.696) and a 64.2% increased risk of gestational hypertension (GH) (OR = 1.642; 95% CI: 1.129–2.389). Those consuming TB four or more times per week faced a 154.3% higher risk of GDM (OR = 2.543; 95% CI: 2.064–3.314) and a 169.3% increased risk of GH (OR = 2.693; 95% CI: 1.773–4.091). Similar results were observed in the analysis of SSB. Regarding offspring health, compared to non-consumers, TB consumption four or more times per week was associated with a substantial increase in the risk of macrosomia (OR = 2.143; 95% CI: 1.304–3.522) and large for gestational age (LGA) (OR = 1.695; 95% CI: 1.219–2.356). In the analysis of NSS, with a significantly increased risk of macrosomia (OR = 6.581; 95% CI:2.796–13.824) and LGA (OR = 7.554; 95% CI: 3.372–16.921). Conclusion: The high level of beverage consumption among pregnant women in Shanghai needs attention. Excessive consumption of beverages increases the risk of GDM and GH, while excessive consumption of NSS possibly has a greater impact on offspring macrosomia and LGA.

## 1. Introduction

The excessive consumption of sugar-sweetened beverages (SSB) has garnered widespread attention globally, emerging as a significant public health challenge. These beverages, available in various flavours, have increasingly gained popularity, leading to rapid growth in their consumption worldwide [1]. This trend has significantly influenced the global beverage market. By 2009, the total consumption of SSB reached approximately 1.6 trillion liters globally, equivalent to an average annual consumption of 231 liters per person [2]. An analysis conducted in 185 countries revealed that by 2018, adults worldwide were consuming approximately 670 g of SSB per week [3]. Recent studies indicate that in 2023, the global average consumption of SSB remains high, with significant variations across different regions. In China, the impact of this global trend is particularly evident. As China’s beverage market continues to expand, beverage annual production has surpassed 180 million tons, a staggering 440-fold increase from 25 years ago [4]. A survey conducted in 27 cities in China in 2016 showed that 74% of children (4–9 years old), 85% of adolescents (10–17 years old), and 83% of adults (18–55 years old) consumed at least 500 mL of SSB per week [5]. Especially, in Shanghai, as the city increasingly adopts a globalised lifestyle, Shanghai’s SSB consumption level is higher than the national average.

A study covering 704 commercial sugary beverages showed that the average free sugar was 8.4 g/100 g, mainly fructose, sucrose, and glucose, which were 3.0 g/100 g, 2.9 g/100 g, and 2.5 g/100 g, respectively [6]. Excessive consumption of SSB can cause various health problems, such as caries, weight gain, and an increased risk of many chronic diseases like diabetes and hypertension [7,8]. Therefore, it is crucial to reduce sugar intake, particularly from beverages, to maintain health. The WHO recommends limiting daily free sugar intake to less than 10% of total energy, ideally less than 5%, to manage health risks associated with the intake of sugar-sweetened beverages [9]. 

The phenomenon of consuming SSB around the globe not only poses particular harm to the general population but may have more severe and far-reaching impacts on pregnant women and their offspring. In general, pregnant women that consume SSB frequently are at risk of developing gestational diabetes mellitus (GDM). As shown in the International Diabetes Federation’s 2021 report, there is a serious risk of developing GDM as a result of high-level beverage consumptions in pregnant women [1,9]. This report also highlighted that GDM is a significant result of maternal hyperglycaemia, affecting many pregnancies. Additionally, research involving 32,933 Norwegian women pregnant for the first time revealed that those with high SSB consumption faced a substantially increased risk of developing preeclampsia, particularly among those with higher intake levels [10]. Excessive consumption of SSB by pregnant women can impact their health and may also result in long-term adverse effects on the foetus, such as premature birth and birth defects, among other poor pregnancy outcomes [11,12,13]. These findings underscore the importance of regulating sugar-sweetened beverage intake during pregnancy to protect the health of both mothers and infants.

In recent years, there has been a significant change in beverage consumption habits and patterns, mainly reflected in two aspects. Firstly, freshly made and sold beverages, such as milk tea, has become extremely popular in China due to their freshness and customizability. This trend is evident from the approximately 515,000 freshly made tea shops in China in 2023 [14]. A study examining 122 varieties of milk tea in Shanghai revealed that a typical full-sugar milk tea contains an average of 7.96 g of sugar per 100 mL [15]. Consequently, the sugar content in a 500 mL cup of milk tea far exceeds the recommended daily intake. Secondly, the consumption of non-sugar sweetened beverages (NSS) has drastically increased, reaching CYN 9.87 billion in 2019 (up from CYN 1.66 billion in 2014), with a compound annual growth rate of 42.84%. It is expected to reach CYN 27.66 billion by 2027 [16]. Previous research has primarily focused on packaged SSB, with relatively little analysis and few data available on freshly made beverages and NSS. Additionally, there is a notable lack of studies on the impact of pregnant women’s intake on the health of their offspring. We hypothesize that high consumption of beverages among pregnant women is associated with increased risks of adverse maternal outcomes and adverse offspring outcomes. To this end, we have launched a comprehensive research project in Shanghai aimed at exploring the potential impacts of these beverages on pregnant women. 

## 2. Method

### 2.1. Participants

This study employed a prospective cohort design, conducting surveys over two consecutive years, 2022 and 2023. Each survey was completed between April and June of the respective year. The sampling methods remained consistent throughout the study. Shanghai has 16 districts. Based on geographical directions, we categorised each region into five zones. One town was randomly selected from each zone, ensuring no overlap in selections between the two years. This yielded a total of 160 towns sampled over the two-year period. From each town, 30 pregnant women were randomly selected, with an equal distribution among the different stages of pregnancy. The study participants were pregnant women living in the community for more than 6 months in last year, who were able to walk independently, had no cognitive impairment, and volunteered to participate in our study. The research process is shown in Figure 1.

### 2.2. Baseline Data Collection 

The questionnaire used in this project was developed after multiple discussions among 5 experts. The investigation employed a 1-on-1 interviewing approach, with a survey questionnaire covering diverse aspects such as general demographic information, including age, education level, marital status, employment status, per capita income, alcohol consumption prior to pregnancy, and so on, the use of nutritional supplements, and beverage consumption frequency. The beverages are divided into eight categories: carbonated beverages (CB), pure fruit juice (PFJ), juice beverages (JB), vegetable protein beverages (VPB), sugar-sweetened dairy and dairy-based beverages (SDB), lactic acid bacteria beverages (LBB), sugar-sweetened tea beverages (including freshly made milk tea beverages) (STB), and NSS. Participants reported their intake frequency of each type of beverage over the past month, choosing from frequency options ranging never, 1–3 times per month, 1–3 times per week, 4–7 times per week, and >1 per day. The average volume consumed were meticulously recorded. 

### 2.3. Follow-Up Data Collection

As of 31 March 2024, all pregnant women have completed their delivery. Follow-up surveys on maternal and offspring outcomes were conducted for pregnant women, excluding those with pre-pregnancy hypertension or diabetes. This information included details on births within the year following the surveys, encompassing outcomes such as gestational diabetes mellitus (GDM), gestational hypertension (GH), miscarriage, and offspring birth weight, length, and gestational age.

### 2.4. Covariates and Categorization

According to the standards of The American College of Obstetricians and Gynaecologists, early pregnancy refers to weeks 1 to 13 weeks and 6 days of pregnancy. The middle pregnancy spans from week 14 weeks and 0 days to 27 weeks and 6 days, and the late pregnancy extends from 28 weeks and 0 days to 40 weeks and 6 days [17]. Family income, referring to the 2022 income situation of residents in Shanghai, was divided into “below average” and “above average”. 

The total beverages (TB) include CB, PFJ, JB, VPB, SDB, LBB, STB, and NSS, whereas SSB include other 7 types of beverages except NSS. The beverage consumption population was defined as those who had consumed beverages at least once in the past month. In the logistic analysis, beverage consumption frequency was divided into 3 categories: no; low-frequency: greater than 0 times/month and less than or equal to 3 times/week; high-frequency: greater than or equal to 4 times/week.

Macrosomia is defined as a birth weight of over 4000 g regardless of gestational age [18]. Similarly, large-gestational age (LGA), which refers to infants whose weight at birth is greater than the 90th percentile for their gestational age, is not only associated with metabolic disorders in later life but also with increased perinatal morbidity [19].

To better understand the impact of birth weight and gestational age on neonatal and maternal health, several key definitions are used in this study. Preterm birth (PTB) is defined as the birth of an infant before 37 weeks of gestation. Low birth weight (LBW) is defined as a birth weight of less than 2500 g regardless of gestational age [20]. Small for gestational age (SGA) refers to infants whose birth weight is below the 10th percentile for their gestational age [21].

### 2.5. Mass Control during Project Implementation

It was conducted by the Shanghai Municipal Centre of Disease Control and Prevention project team, which initiated training sessions for personnel from various district disease control departments. Successful completion of the training and relevant assessments was a prerequisite for the participants to assume their roles. Trained personnel conducted surveys and collected data to minimize errors and reduce recall bias. The project team also regularly monitored the data collection process to ensure adherence to study protocols and address any issues that arose.

By implementing this structured and detailed mass control process, the study was able to accurately analyze the correlations between the consumption of SSBs and NSSs and adverse pregnancy outcomes. This approach ensured the reliability and validity of the findings, highlighting the significant public health implications of beverage consumption among pregnant women in Shanghai.

### 2.6. Statistical Analysis

Statistical analysis was conducted using SPSS version 25.0. All tests were two-sided, with *p* < 0.05 indicating statistical significance. The chi-square test was used to analyse qualitative variables, while non-parametric tests were used for quantitative variables. The chi-square test evaluated the composition ratio of beverage consumption. Logistic regression analysis was performed to assess the impact of different type beverage consumption, along with other potential influencing factors including age, education level, income, employment status, alcohol consumption outside of pregnancy, use of nutritional supplements, and BMI before pregnancy, on various adverse maternal and offspring outcomes.

## 3. Results

### 3.1. Socio-Demographic Profile and Health Behaviours of Pregnant Women across Different Pregnancy Stages

A survey of 4824 pregnant women was completed, with 82.2% of these participants being under the age of 35 in Table 1. Analysis of the results indicated significant statistical differences across different pregnancy stages in terms of age, per capita income, alcohol consumption prior to pregnancy, and nutritional supplement intake (*p* < 0.05). The other investigated factors did not exhibit statistically significant differences. 

### 3.2. Frequency and Volume of Consumption of Different Types of Beverages

The consumption rates of TB, SSB, and NSS among pregnant women in Shanghai were 73.2%, 72.8%, and 13.5%, respectively. The rates of different consumption frequencies are shown in Table 2. Significant statistical differences were observed in the composition ratios of TB and PFJ consumption across different stages of pregnancy (*p* < 0.05). The median consumption volumes of TB, SSB, and NSS among the consumer group were 66.7 mL, 65.0 mL, and 16.7 mL, respectively, with no statistical differences across different pregnancy stages.

In the analysis of the composition of beverage consumption, it was found that SSB are the predominant type, accounting for 94.2% of the TB consumption. Within the SSB category, the top three products by consumption share are SDB, PFJ, and STB, which account for 25.3%, 18.9%, and 13.4% of TB consumption, respectively in Figure 2. 

### 3.3. Beverage Consumption Effect of Pregnancy Outcomes and Offspring Health

We tracked the pregnancy outcomes of 4635 women and offspring health outcomes of 4000 women. The follow-up rates were 96.1% and 83.0% in Table 3. The incidence rates were 16.9% for GDM and 4.9% for GH. In offspring health outcomes, the incidence rates were 8.4% for miscarriage, 5.2% for PTB, 3.8% for LBW, 4.8% for macrosomia, 8.0% for SGA, and 12.1% for LGA. 

Further analysis revealed that the incidence rate of GDM in the group consuming TB was 19.6%, compared to 12.2% in non-consumers, 4.9% versus 2.5% for GH, and 12.8% versus 10% for LGA. Delving into the specifics of SSB consumption, we found a GDM incidence rate of 19.6% in consumers versus 12.3% in non-consumers, a GH rate of 4.9% compared to 2.5%, a LGA rate of 12.9% versus 10.0%, and a birth weight of 3274 g + 445 compared to 3234 g ± 500. Additionally, when comparing those who consumed NSS beverage to non-consumers, the incidence rates were 22.8% versus 16.8% for GDM, 6.6% versus 3.9% for hypertension, and 7.8% versus 4.3% for macrosomia. All the above differences have been statistically significant (*p* < 0.05).

### 3.4. Logistic Analysis of the Relationship between Beverage Consumption Frequency and the Risk of Adverse Maternal and Offspring Outcomes

Logistic regression analysis was performed to assess the impact of TB consumption, along with other potential influencing factors including age, education level, income, employment status, alcohol consumption outside of pregnancy, use of nutritional supplements, and BMI before pregnancy, on various adverse maternal outcomes. When analyzing factors affecting offspring outcomes, GDM was included as a dependent variable in addition to the above factors. The results indicated that, compared to non-consumers, pregnant women with low-frequency TB consumption experienced a 38.4% increased risk of GDM (OR = 1.384; 95% CI: 1.129–1.696) and a 64.2% increased risk of GH (OR = 1.642; 95% CI: 1.129–2.389). Those with high-frequency TB consumption faced a significantly higher risk, with a 154.3% increase for GDM (OR = 2.543; 95% CI: 2.064–3.314) and a 169.3% increase for GH (OR = 2.693; 95% CI: 1.773–4.091). Regarding offspring health, TB consumption did not significantly affect the risk of macrosomia, PTB, LBW, and SGA. However, pregnant women with high-frequency TB consumption were associated with a substantial increase in the risk of macrosomia (OR = 2.143; 95% CI: 1.304–3.522) and LGA (OR = 1.695; 95% CI: 1.219–2.356), as detailed in Table 4.

The same analysis method was used to analyze the impact of SSB consumption on outcomes. After adjusting for confounding factors, it was found that pregnant women with low-frequency SSB consumption had a 47.8% increased risk of GDM (OR = 1.478; 95% CI: 1.199–1.822). Those with high-frequency SSB consumption faced a 157.8% higher risk (OR = 2.578; 95% CI: 2.064–3.222). A similar pattern was observed for GH; compared to non-consumers, low-frequency SSB consumers faced a 78.9% increased risk (OR = 1.789; 95% CI: 1.164–2.75), and high-frequency consumers faced a 179.7% increased risk (OR = 2.797; 95% CI: 1.788–4.376). Regarding offspring health, pregnant women with high-frequency SSB consumption were associated with a substantial increase in the risk of LGA (OR = 1.476; 95% CI: 1.041–2.094), as detailed in Table 5.

The same analysis method was used to analyze the impact of NSS consumption on outcomes. It showed that, compared to non-consumers, pregnant women with high-frequency NSS consumption were associated with a substantial increase in the risk of macrosomia (OR = 6.581; 95% CI: 2.796–13.824) and LGA (OR = 7.554; 95% CI: 3.372–16.921), as detailed in Table 6.

## 4. Discussion

As beverage consumption continues to rise in China, so does the intake among pregnant women, indicating a troubling trend. According to our study, the consumption rates of TB, SSB, and NSS among pregnant women in Shanghai were 73.2%, 72.8%, and13.5%, respectively. The median consumption volumes TB, SSB, NSS among the consumer group were 66.7 mL, 65.0 mL, and 16.7 mL, respectively. According to the 2017 Behavioral Risk Factor Surveillance System by the CDC, over one-fifth of American pregnant women consume SSB at least once daily [22]. Our study revealed that 11.8% of pregnant women reported daily consumption of sugary beverages.

Although the daily consumption rate of sugary beverages among Shanghai pregnant women is lower than that in the U.S., the substantial consumer base and significant volume of consumption still necessitate close attention. Notably, the beverage choices between pregnant women and the general adult population differ significantly. Adults tend to favor CB and milk tea, whereas pregnant women more frequently opt for SDB, PFJ, and STB [23]. Even though these beverages appear healthier, they often contain considerable amounts of added sugars, the potential health impacts of which should not be overlooked.

This study demonstrates that frequent consumption of sugary beverages is strongly linked to increased risks of GDM and GH among pregnant women. It specifically reveals that women consuming sugary beverages four or more times per week are a considerably higher risk of developing GDM (OR = 2.543) and GH (OR = 2.693). These findings are consistent with those from the Spanish SUN project, which also reported a significant correlation between high consumption of sugary soft drinks before pregnancy and the onset of GDM (OR = 3.06) [24]. The analysis in Japan found that women who consume sugary cola five or more times a week had a 22% higher risk of developing GDM compared to those who consume less than one serving per month, highlighting the influence of dietary habits on GDM risk though this study did not consider juice consumption [25]. Moreover, a study in Brazil found that among 1370 pregnant women, 14.0% had gestational hypertension, and 30.4% of them consumed soft drinks seven or more times per week [26]. 

The consumption of TB, SSB, and NSS beverages showed significant correlations with the occurrence of macrosomia and LGA infants. Pregnant women who consumed NSS four or more times per week face significantly increased risks macrosomia (OR = 6.581) and LGA (OR = 7.554). Although NSS beverages generally contain fewer calories than SSB, research has shown that both types of drinks associated with similar adverse health outcomes, indicating a significant impact on fetal growth [27]. This suggests a dose-response relationship between beverage consumption frequency and the severity of these outcomes. These findings align with research indicating a consistent association between high sugar intake during pregnancy and increased birth weight, potentially leading to complications like macrosomia and LGA [28,29,30]. A meta-analysis highlighted that maternal sugar consumption significantly correlates with a higher risk of delivering LGA infants [12]. To better understand these adverse outcomes, it is essential to consider the underlying biological mechanisms. Frequent consumption of SSBs can lead to increased glucose levels and insulin resistance, significant risk factors for GDM and GH. High sugar intake during pregnancy can elevate maternal blood glucose levels, increasing fetal insulin production. Insulin acts as a growth factor for the fetus, leading to macrosomia and LGA. Although NSS beverages contain fewer calories, they can still disrupt gut microbiota and metabolic processes, potentially leading to glucose intolerance and insulin resistance. High consumption of SSBs and NSS can displace more nutrient-dense food and beverages from the diet, leading to nutritional deficiencies, exacerbating the risk of GDM, GH, and poor fetal growth [31]. Given these insights, it is crucial for healthcare providers to recommend dietary modifications that limit the intake of both SSB and NSS beverages during pregnancy to mitigate these risks. 

Our statistical analysis found no significant associations between the consumption of SSB and miscarriage, LBW, and SGA This is consistent with findings from a large U.S. study using NHANES data, which also showed no direct association between the consumption of sugar-sweetened beverages and increased rates of preterm births [32]. Additionally, a study found that ordinary beverage consumption did not significantly impact miscarriage rates [33]. A multinational study evaluated the relationship between beverage consumption and the risk of low birth weight (LBW) and small-for-gestational-age (SGA) infants. The results showed that regular beverage consumption did not significantly increase the risk of LBW or SGA [34,35]. While some studies suggest a potential link between sugary drink consumption and an increased risk of preterm birth, larger-scale studies have demonstrated that regular beverage consumption does not significantly affect preterm birth rates [35,36,37].

However, the analysis showed TB may have potential protective factors against PTB, possibly due to their energy content. However, while sugary drinks provide a quick energy boost, they are nutritionally deficient and contribute to an imbalanced diet. Thus, it is not recommended to use sugary beverages as a strategy to prevent PTB. Instead, it is essential to advocate for more stringent dietary guidelines for pregnant women to mitigate risks [38]. Proactive beverage management can improve maternal health outcomes and reduce the likelihood of complications during pregnancy. For pregnant women, educational strategies should be particularly empathetic and supportive, offering practical and accessible information [39]. Health education can be effectively delivered through prenatal classes that include nutrition counseling, where pregnant women can learn about the importance of balanced diets and the specific risks associated with excessive consumption of certain types of beverages [40,41]. Additionally, digital platforms like specific apps can provide daily tips and trackers for food and beverage intake, helping women to monitor and adjust their consumption habits in real-time [40]. 

Our findings have significant clinical implications for the management of maternal and fetal health. Clinicians should be aware of the potential risks associated with both SSBs and NSS, and provide comprehensive beverage and dietary counseling to pregnant women. Implementing these findings into clinical practice can help in the early identification and management of at-risk pregnancies, potentially reducing the incidence of GDM, GH, macrosomia, and LGA. Public health initiatives can also be designed to educate women of childbearing age about the potential risks of excessive consumption of both SSBs and NSS during pregnancy.

Our multistage sampling method ensures sample representativeness and reliability, allowing us to draw meaningful conclusions from a diverse population. By analyzing different beverage types and eight adverse pregnancy outcomes, our study provides a comprehensive perspective. This breadth enhances the robustness of our findings and offers a nuanced understanding of the relationships between beverage consumption and pregnancy outcomes. One key strength of our study is the large sample size, which increases the statistical power and precision of our estimates. Additionally, our study addresses a significant gap in the literature by examining both traditional and newly popular beverage types, including freshly made and non-sugar sweetened beverages, which have seen a rise in consumption but lack substantial research. However, there are limitations to consider. We did not account for potential confounders such as overall diet, physical activity levels, and genetic predispositions, which could introduce residual confounding. The cross-sectional design limits our ability to infer causality, as it captures data at a single point in time. Furthermore, despite efforts to minimize errors, data collection through survey questionnaires may still be susceptible to recall bias and social desirability bias.

## 5. Conclusions

In summary, our study shows that while the beverage consumption rate among pregnant women in Shanghai remains significant at 73.2%. Excessive consumption of beverages is linked to increased incidences of GDM and GH, as well as higher occurrences of macrosomia and LGA infants. This trend is particularly pronounced with NSS beverage consumption. Therefore, it is crucial to enhance health education regarding beverage intake during pregnancy, guiding pregnant women towards reasonable dietary choices to promote both maternal and offspring health.

## Figures and Tables

**Figure 1 nutrients-16-02412-f001:**
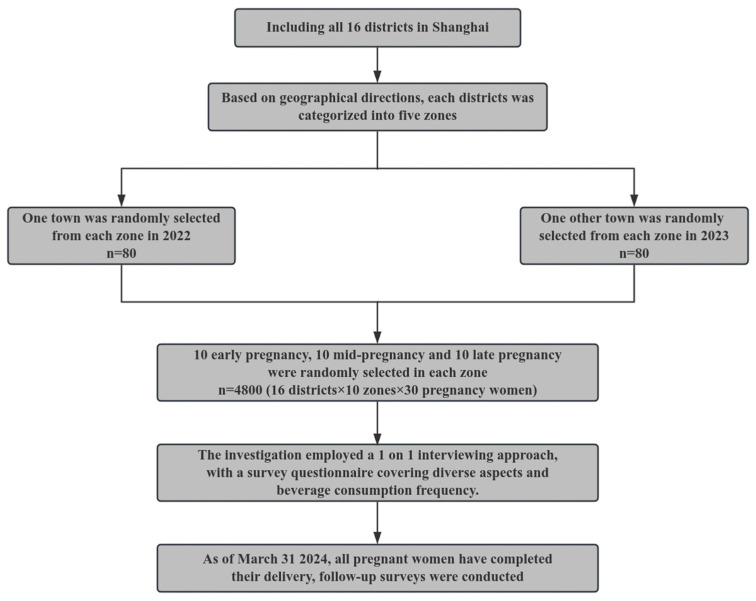
The project flowchart.

**Figure 2 nutrients-16-02412-f002:**
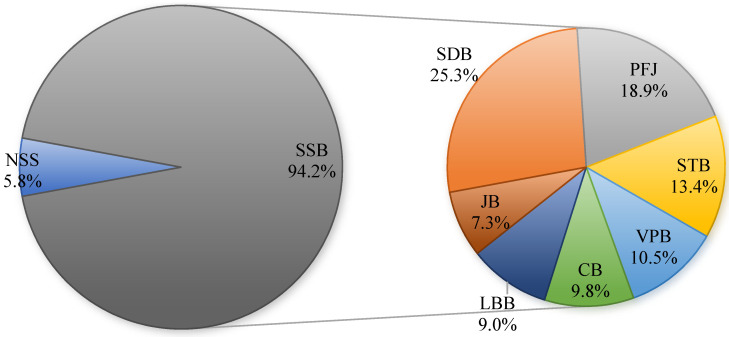
Consumption composition of different beverage.

**Table 1 nutrients-16-02412-t001:** Socio-demographic profile and health behaviours of pregnant women across different stages of pregnancy.

	Total	Early Pregnancy	Mid-Pregnancy	Late Pregnancy	*p*
Characteristics	N (%)	N (%)	N (%)	N (%)	
Total numble	4824	1638	1618	1568	/*
Age					
<35	3966 (82.2)	1370 (83.6)	1339 (82.8)	1257 (80.2)	<0.05
≥35	858 (17.8)	268 (16.4)	279 (17.2)	311 (19.8)	
Education Level					
Specialty or lower	2012 (41.7)	661 (40.4)	688 (42.5)	663 (42.3)	0.587
Undergraduate	2162 (44.8)	741 (45.2)	719 (44.4)	702 (44.8)	
Postgraduate	650 (13.5)	236 (14.4)	211 (13.0)	203 (12.9)	
Marital status					
Unmarried, divorced or separated	57 (1.2)	26 (1.6)	19 (1.2)	12 (0.8)	0.098
Married or cohabiting	4767 (98.8)	1612 (98.4)	1599 (98.8)	1556 (99.2)	
Employment status					
Mental Labour	2328 (48.3)	800 (48.8)	747 (46.2)	781 (49.8)	0.128
Physical Labour	912 (18.9)	323 (19.7)	316 (19.5)	273 (17.4)	
Others	1584 (32.8)	515 (31.4)	555 (34.3)	514 (32.8)	
Per capita income					
Below average	2368 (49.1)	755 (46.1)	817 (50.5)	796 (50.8)	0.012
Over average	2456 (50.9)	883 (53.9)	801 (49.5)	772 (49.2)	
Alcohol consumption prior to pregnancy					
Yes	319 (6.6)	133 (8.1)	95 (5.9)	91 (5.8)	0.010
No	4505 (93.4)	1505 (91.9)	1523 (94.1)	1477 (94.2)	
Taking nutritional supplement					
Yes	3085 (64.0)	965 (58.9)	1077 (66.6)	1043 (66.5)	<0.001
No	1739 (36.0)	673 (41.1)	541 (33.4)	525 (33.5)	
BMI Before pregnancy					
Underweight	477 (9.9)	157 (9.6)	165 (10.2)	155 (9.9)	0.473
Normal weight	3273 (67.8)	1092 (66.7)	1100 (68.0)	1081 (68.9)	
Overweight and obesity	1074 (22.3)	389 (23.7)	353 (21.8)	332 (21.2)	

/*: No statistical testing was conducted.

**Table 2 nutrients-16-02412-t002:** Frequency of consumption of different types of sugary beverages.

	Consumption Frequency (%)					*p*	Consumption Volume (mL) Median (P25, P75)	*p*
Characteristic	Never	1–3 Times Per Month	1–3 Times Per Week	4–7 Times Per Week	>1 Per Day			
TB	26.8	10.4	32.6	17.6	12.5		66.7 (26.7, 146.7)	
Early pregnancy	24.8	10.6	34.9	17.5	12.1	0.034	66.7 (26.7, 140.0)	0.755
Mid-pregnancy	26.6	9.3	33.3	18.1	12.7		66.7 (30.0, 140.0)	
Late pregnancy	29.0	11.5	29.6	17.2	12.8		66.7 (26.7, 150.0)	
SSB	27.2	11.2	32.7	17.1	11.8		65.0 (26.7, 133.3)	
Early pregnancy	25.4	11.2	35.2	16.7	11.5	0.052	60.0 (26.7, 133.3)	0.702
Mid-pregnancy	26.8	10.3	33.3	17.6	12.1		66.7 (26.7, 133.3)	
Late pregnancy	29.4	12.0	29.6	17.1	11.9		60.0 (26.7, 146.7)	
CB	70.7	23.2	5.0	1.0	0.2		13.3 (6.7, 26.7)	
Early pregnancy	70.6	24.1	4.3	0.9	0.2	0.127	13.3 (8.0, 26.7)	0.166
Mid-pregnancy	69.1	23.6	5.9	1.3	0.1		16.7 (10.0, 26.7)	
Late pregnancy	72.4	21.9	4.8	0.7	0.3		13.3 (6.7, 26.7)	
PFJ	59.1	28.7	9.4	2.2	0.6		16.7 (13.3, 40.0)	
Early pregnancy	56.6	31.9	8.9	2.0	0.7	0.008	16.7 (13.3, 33.3)	0.831
Mid-pregnancy	59.1	27.8	10.6	1.9	0.6		16.7 (13.3, 53.3)	
Late pregnancy	61.7	26.2	8.7	2.8	0.6		16.7 (13.3, 53.3)	
FJ	78.9	16.9	3.0	1.0	0.3		13.3 (6.7, 26.7)	
Early pregnancy	77.8	17.7	3.1	1.1	0.4	0.784	13.3 (6.7, 33.3)	0.202
Mid-pregnancy	79.7	16.1	3.0	0.8	0.4		13.3 (6.7, 25.0)	
Late pregnancy	79.2	16.7	2.9	1.1	0.1		13.3 (6.7, 20.0)	
VPB	79.0	13.0	5.4	1.9	0.7		16.7 (13.3, 53.3)	
Early pregnancy	77.5	14.1	5.6	1.8	1.0	0.059	16.7 (13.3, 53.3)	0.108
Mid-pregnancy	79.1	11.9	6.2	2.3	0.6		16.7 (13.3, 53.3)	
Late pregnancy	80.6	12.9	4.3	1.7	0.5		16.7 (13.3, 50.0)	
SDB	61.0	16.4	14.0	6.2	2.4		26.7 (13.3, 66.7)	
Early pregnancy	59.5	18.3	113.9	5.8	2.5	0.124	26.7 (13.3, 66.7)	0.105
Mid-pregnancy	61.5	15.9	14.8	5.7	2.1		26.7 (13.3, 66.7)	
Late pregnancy	61.9	15.1	13.1	7.1	2.7		26.7 (13.3, 76.7)	
LBB	72.5	17.4	7.7	1.7	0.7		13.3 (6.7, 26.7)	
Early pregnancy	72.5	18.3	7.1	1.5	0.7	0.325	13.3 (6.7, 26.7)	0.314
Mid-pregnancy	71.8	18.2	7.8	1.4	0.8		13.3 (6.7, 26.7)	
Late pregnancy	73.2	15.8	8.2	2.2	0.7		13.3 (6.7, 26.7)	
STB	68.3	24.1	6.7	0.7	0.2		20.0 (13.3, 33.3)	
Early pregnancy	67.3	24.1	7.4	0.9	0.4	0.253	20.0 (13.3, 33.3)	0.494
Mid-pregnancy	69.7	23.2	6.3	0.7	0.1		20.0 (13.3, 33.3)	
Late pregnancy	68.1	24.9	6.3	0.6	0.1		20.0 (13.3, 33.3)	
NSS	86.5	10.3	2.5	0.3	0.4		16.7 (10.0, 33.3)	
Early pregnancy	84.9	11.8	2.6	0.2	0.5	0.119	13.3 (6.7, 33.3)	0.297
Mid-pregnancy	87.1	10.0	2.1	0.5	0.3		16.7 (10.0, 33.3)	
Late pregnancy	87.7	9.0	2.8	0.3	0.3		20.0 (13.3, 33.3)	

**Table 3 nutrients-16-02412-t003:** Impact of beverage consumption on pregnancy outcomes and offspring health.

Factors	GDM		GH		Miscarriage		PTB		LBW		Macrosomia		SGA		LGA		Birth Weight	
	N (%)	*p*	N (%)	*p*	N (%)	*p*	N (%)	*p*	N (%)	*p*	N (%)	*p*	N (%)	*p*	N (%)	*p*	g (Mean ± SD)	*p*
Total	815 (17.6)		197 (4.2)		231 (8.4)		189 (5.2)		140 (3.8)		175 (4.8)		291 (8.0)		441 (12.1)		3264 ± 461	
TB (SSB and NSS)																		
Yes	664 (19.6)	<0.001	166 (4.9)	<0.001	176 (8.6)	0.488	125 (4.6)	0.039	91 (3.5)	0.118	140 (5.2)	0.106	207 (7.7)	0.297	347 (12.8)	0.043	3236 ± 500	0.066
No	151 (12.2)		31 (2.5)		55 (7.8)		64 (6.8)		49 (4.9)		35 (3.7)		84 (9.0)		94 (10.0)		3274 ± 446	
SSB																		
Yes	660 (19.6)	<0.001	165 (4.9)	<0.001	176 (8.7)	0.391	125 (4.6)	0.052	91 (3.4)	0.064	139 (5.1)	0.130	203 (7.5)	0.181	346 (12.9)	0.036	3234 ± 500	0.046
No	155 (12.3)		32 (2.5)		55 (7.7)		64 (6.7)		49 (5.1)		36 (3.8)		88 (9.2)		95 (10.0)		3274 + 445	
NSS																		
Yes	142 (22.8)	0.001	41 (6.6)	0.008	30 (7.2)	0.334	21 (3.8)	0.123	14 (2.5)	0.078	43 (7.8)	0.013	43 (7.8)	0.872	84 (15.1)	0.068	3259 ± 453	0.169
No	673 (16.8)		156 (3.9)		201 (8.6)		168 (5.4)		126 (4.1)		132 (4.3)		248 (8.0)		457 (11.6)		3294 ± 502	

**Table 4 nutrients-16-02412-t004:** Logistic analysis of the relationship between TB consumption frequency and the risk of adverse maternal and offspring outcomes.

TB Consumption Frequency	Total			Early Pregnancy			Mid- Pregnancy			Late Pregnancy		
	OR	*p*	95% CI	OR	*p*	95% CI	OR	*p*	95%CI	OR	*p*	95%CI
GDM												
0	Reference			Reference			Reference			Reference		
low-frequency	1.384	0.002	1.129–1.696	1.642	0.010	1.129–2.389	1.965	<0.001	1.348–2.832	0.923	0.0640	0.661–1.291
high-frequency	2.543	<0.001	2.064–3.314	3.693	<0.001	2.516–5.421	2.811	<0.001	1.909–4.138	1.748	<0.001	1.246–2.452
GH												
0	Reference			Reference			Reference			Reference		
low-frequency	1.706	0.011	1.130–2.575	2.201	0.088	0.889–5.450	2.878	0.008	1.321–6.273	1.056	0.860	0.575–1.942
high-frequency	2.693	<0.001	1.773–4.091	4.117	0.002	1.656–10.232	3.150	0.006	1.399–7.092	2.010	0.022	1.108–3.648
Miscarriage												
0	Reference			Reference			Reference			Reference		
low-frequency	1.202	0.288	0.856–1.689	1.395	0.231	0.809–2.405	0.830	0.521	0.469–1.467	1.189	0.646	0.568–2.492
high-frequency	0.836	0.369	0.566–1.235	0.848	0.611	0.448–1.604	0.727	0.329	0.383–1.379	0.919	0.839	0.404–2.088
PTB												
0	Reference			Reference			Reference			Reference		
low-frequency	0.683	0.078	0.447–1.043	0.544	0.086	0.272–1.089	0.723	0.397	0.342–1.530	0.752	0.475	0.344–1.644
high-frequency	0.816	0.381	0.517–1.287	0.580	0.181	0.261–1.288	0.837	0.671	0.369–1.900	1.065	0.875	0.486–2.335
LBW												
0	Reference			Reference			Reference			Reference		
low-frequency	0.714	0.166	0.443–1.151	1.128	0.774	0.496–2.565	0.538	0.233	0.194–1.490	0.525	0.107	0.240–1.149
high-frequency	0.690	0.177	0.403–1.182	1.307	0.562	0.528–3.235	0.521	0.276	0.162–1.682	0.435	0.068	0.177–1.064
Macrosomia												
0	Reference			Reference			Reference			Reference		
low-frequency	0.961	0.879	0.573–1.610	3.791	0.080	0.852–16.875	0.489	0.114	0.202–1.187	0.891	0.777	0.401–1.981
high-frequency	2.143	0.003	1.304–3.522	10.063	0.002	2.296–44.104	1.156	0.734	0.500–2.673	1.819	0.129	0.839–3.941
SGA												
0	Reference			Reference			Reference			Reference		
low-frequency	0.821	0.266	0.581–1.162	0.633	0.154	0.337–1.187	0.998	0.995	0.527–1.891	0.812	0.467	0.464–1.423
high-frequency	0.786	0.224	0.533–1.159	0.74	0.394	0.371–1.478	0.552	0.149	0.246–1.237	0.952	0.870	0.527–1.720
LGA												
0	Reference			Reference			Reference			Reference		
low-frequency	1.160	0.36	0.844–1.593	3.26	0.009	1.339–7.938	0.979	0.938	0.568–1.685	0.922	0.734	0.577–1.473
high-frequency	1.695	0.002	1.219–2.356	7.153	<0.001	2.917–17.539	1.134	0.674	0.631–2.036	1.205	0.459	0.736–1.972

**Table 5 nutrients-16-02412-t005:** Logistic analysis of the relationship between SSB consumption frequency and the risk of adverse maternal and offspring outcomes.

SSB Consumption Frequency	Total			Early Pregnancy			Mid- Pregnancy			Late Pregnancy		
	OR	*p*	95% CI	OR	*p*	95% CI	OR	*p*	95%CI	OR	*p*	95%CI
GDM												
0	Reference			Reference			Reference			Reference		
low-frequency	1.478	<0.001	1.199–1.822	1.788	0.003	1.221–2.618	2.094	<0.001	1.411–3.107	0.984	0.928	0.700–1.383
high-frequency	2.578	<0.001	2.064–3.222	3.450	<0.001	2.296–5.184	3.275	<0.001	2.160–4.965	1.756	0.002	1.225–2.516
GH												
0	Reference			Reference			Reference			Reference		
low-frequency	1.789	0.008	1.164–2.750	2.852	0.034	1.081–7.524	2.738	0.018	1.189–6.306	1.062	0.851	0.568–1.987
high-frequency	2.797	<0.001	1.788–4.376	4.218	0.006	1.525–11.665	3.004	0.014	1.245–7.249	2.381	0.006	1.278–4.434
Miscarriage												
0	Reference			Reference			Reference			Reference		
low-frequency	1.234	0.229	0.876–1.738	1.432	0.198	0.829–2.475	0.922	0.782	0.520–1.634	1.082	0.837	0.510–2.297
high-frequency	0.944	0.781	0.632–1.412	0.94	0.854	0.485–1.822	0.854	0.637	0.443–1.645	1.030	0.946	0.445–2.383
PTB												
0	Reference			Reference			Reference			Reference		
low-frequency	0.728	0.141	0.476–1.112	0.578	0.124	0.288–1.162	0.734	0.423	0.344–1.565	0.843	0.664	0.390–1.824
high-frequency	0.885	0.615	0.550–1.423	0.808	0.609	0.358–1.827	0.813	0.636	0.345–1.918	1.079	0.857	0.472–2.468
LBW												
0	Reference			Reference			Reference			Reference		
low-frequency	0.669	0.100	0.415–1.080	1.044	0.916	0.469–2.323	0.386	0.076	0.135–1.104	0.569	0.160	0.259–1.251
high-frequency	0.767	0.343	0.444–1.326	1.413	0.459	0.565–3.533	0.489	0.232	0.151–1.582	0.612	0.287	0.248–1.511
Macrosomia												
0	Reference			Reference			Reference			Reference		
low-frequency	1.060	0.823	0.636–1.766	4.234	0.057	0.957–18.728	0.548	0.180	0.227–1.320	0.991	0.982	0.453–2.169
high-frequency	1.678	0.057	0.984–2.861	8.278	0.006	1.821–37.627	0.819	0.678	0.320–2.100	1.484	0.351	0.647–3.404
SGA												
0	Reference			Reference			Reference			Reference		
low-frequency	0.740	0.091	0.522–1.049	0.576	0.087	0.307–1.083	0.779	0.444	0.412–1.475	0.822	0.497	0.468–1.445
high-frequency	0.814	0.310	0.547–1.211	0.766	0.465	0.375–1.566	0.523	0.116	0.233–1.173	1.095	0.772	0.593–2.020
LGA												
0	Reference			Reference			Reference			Reference		
low-frequency	1.250	0.168	0.910–1.716	3.777	0.003	1.558–9.156	1.02	0.944	0.589–1.768	0.981	0.936	0.615–1.564
high-frequency	1.476	0.029	1.041–2.094	6.02	<0.001	2.387–15.182	1.002	0.996	0.538–1.867	1.084	0.764	0.641–1.831

**Table 6 nutrients-16-02412-t006:** Logistic analysis of the relationship between NSS consumption frequency and the risk of adverse maternal and offspring outcomes.

NSS Consumption Frequency	Total			Early Pregnancy			Mid- Pregnancy			Late Pregnancy		
	OR	*p*	95% CI	OR	*p*	95% CI	OR	*p*	95%CI	OR	*p*	95%CI
GDM												
0	Reference			Reference			Reference			Reference		
low-frequency	1.181	0.134	0.950–1.469	1.178	0.365	0.826–1.680	1.176	0.409	0.800–1.728	1.18	0.424	0.787–1.770
high-frequency	1.771	0.130	0.844–3.712	3.970	0.022	1.222–12.895	0.599	0.511	0.130–2.76	2.123	0.317	0.486–9.285
GH												
0	Reference			Reference			Reference			Reference		
low-frequency	1.283	0.201	0.875–1.880	1.057	0.886	0.495–2.256	1.828	0.052	0.996–3.356	1.028	0.936	0.527–2.005
high-frequency	2.381	0.115	0.810–6.997	4.791	0.057	0.957–23.996	1.571	0.675	0.191–12.923	1.264	0.834	0.141–11.333
Miscarriage												
0	Reference			Reference			Reference			Reference		
low-frequency	0.808	0.315	0.533–1.225	0.869	0.647	0.477–1.584	0.515	0.12	0.224–1.188	1.207	0.66	0.522–2.788
high-frequency	0.365	0.329	0.048–2.760	0.68	0.719	0.083–5.585	/*	/*	/*	/*	/*	/*
PTB												
0	Reference			Reference			Reference			Reference		
low-frequency	0.838	0.544	0.474–1.483	0.516	0.231	0.175–1.524	1.231	0.663	0.484–3.133	0.827	0.71	0.303–2.253
high-frequency	/*	/*	/*	/*	/*	/*	/*	/*	/*	/*	/*	/*
LBW												
0	Reference			Reference			Reference			Reference		
low-frequency	0.655	0.246	0.320–1.339	0.67	0.43	0.248–1.812	1.721	0.425	0.454–6.526	0.195	0.114	0.026–1.481
high-frequency	/*	/*	/*	/*	/*	/*	/*	/*	/*	/*	/*	/*
Macrosomia												
0	Reference			Reference			Reference			Reference		
low-frequency	1.122	0.674	0.657–1.917	1.181	0.699	0.508–2.743	0.562	0.448	0.127–2.486	1.459	0.37	0.639–3.330
high-frequency	6.581	<0.001	2.796–13.824	6.191	0.002	2.505–12.478	17.924	<0.001	4.188–74.016	6.427	0.117	0.629–65.638
SGA												
0	Reference			Reference			Reference			Reference		
low-frequency	1.050	0.822	0.685–1.609	1.097	0.802	0.532–2.265	1.257	0.580	0.558–2.833	0.819	0.585	0.399–1.679
high-frequency	0.446	0.434	0.059–3.364	0.992	0.994	0.118–8.332	/*	/*	/*	/*	/*	/*
LGA												
0	Reference			Reference			Reference			Reference		
low-frequency	1.031	0.866	0.726–1.464	1.052	0.871	0.571–1.936	0.725	0.398	0.344–1.529	1.223	0.469	0.709–2.109
high-frequency	7.554	<0.001	3.372–16.921	7.253	0.002	2.010–26.171	16.746	<0.001	3.847–72.893	1.759	0.617	0.192–16.145

/*: Insufficient data volume for statistical analysis.

## Data Availability

Please contact author for data requests.

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
