# Peer review of "Association of Beverage Consumption during Pregnancy with Adverse Maternal and Offspring Outcomes"

_nutrients, 2024, doi:10.3390/nu16152412_

Round 1

Reviewer 1 Report

Comments and Suggestions for Authors

Dear Authors,

I have now completed the review of the manuscript entitled "Association of beverage consumption during pregnancy with adverse maternal and offspring outcomes". The manuscript is robust, relevant and generally quite well written. However, I have some suggestions to further improve the quality of the manuscript. I would like to suggest that you address the completion of the data in the Materials and Methods - sample selection, description of the questionnaire and its validation/checking in the study group, and data visualisation. I would also suggest that you address these limitations in the article, either by discussing them in the limitations section or, if possible, by making the appropriate revisions:

1. Introduction
 Could you please provide more recent data on the consumption of sugar-sweetened beverages in China and around the world.
For WHO recommendations, use: Guideline: Sugars Intake for Adults (https://www.who.int/publications/i/item/9789241549028) and Children and file:///C:/Users/p160805/Downloads/WHO_carbohydrate_2023.pdf

2. Method
I would like to propose: Materials and methods.
In subsection 2.1 Participants, I suggest describing the group selection in more detail, including group size and inclusion and exclusion factors. Perhaps it would be worth presenting this in a flowchart: study design and data collection.
Subsection 2.2 - the questionnaire should be described in detail, including the question and frequency category of the responses.
Validation: Provide more details on the validation process of the questionnaire used to ensure that the responses are representative and accurate.
 Subsection 2.3. add the year of the study and whether it applies equally to women who participated in the study in 2022 and 2023.
 Subsection 2.3.
The exact classification of the beverages should be provided.
The paragraph on macrosomia does not fit in this chapter - what you wanted to show - explain it.
 Section 2.5 - this section is not very clear to me and needs clarification on what mass control is and how the research was done.
 Sub-section 2.6
The logistic regression analysis needs more description - were any confounding factors taken into account - what were they?

3. Results.
Data visualisation: Use more visual and less text-heavy graphs and tables to make the results clearer and easier to understand.
 In Table 1, add N in the first row - header; are the data needed for x2 - consider whether it is better to delete these values.
Explanations:

*low-frequency: Greater than 0 time/month and less than or equal to 3times/week, high-frequency: greater than or equal to 4 times/week - should also be explained in the methodology.

The statement "The consumption rates of TB, SSB, NSS were as follows: 73.2%, 72.8% and 13.5%". - is too general or is it combined data for each consumption - the frequency of consumption should be analysed, especially daily.

4. Discussion

I suggest that the manuscript be strengthened by paying more attention to the strengths and limitations of the study.

In addition, reference citations should be corrected according to the Guidelines for Authors. Please pay attention to item number 3 - a combination of 2 references.

Summary of Key Findings: Include a visual summary or infographic highlighting the study’s key findings to facilitate quick understanding of the most important points.

Author Response

1.Introduction

 Could you please provide more recent data on the consumption of sugar-sweetened beverages in China and around the world.

For WHO recommendations, use: Guideline: Sugars Intake for Adults (https://www.who.int/publications/i/item/9789241549028) and Children and file:///C:/Users/p160805/Downloads/WHO_carbohydrate_2023.pdf

REPLY:Thank you very much for your suggestion. We have revised the relevant content and references. in

2.Method

I would like to propose: Materials and methods.

In subsection 2.1 Participants, I suggest describing the group selection in more detail, including group size and inclusion and exclusion factors. Perhaps it would be worth presenting this in a flowchart: study design and data collection.

Subsection 2.2 - the questionnaire should be described in detail, including the question and frequency category of the responses.

Validation: Provide more details on the validation process of the questionnaire used to ensure that the responses are representative and accurate.

Subsection 2.3. add the year of the study and whether it applies equally to women who participated in the study in 2022 and 2023.

Subsection 2.4.

The exact classification of the beverages should be provided.

The paragraph on macrosomia does not fit in this chapter - what you wanted to show - explain it.

Section 2.5 - this section is not very clear to me and needs clarification on what mass control is and how the research was done.

 Sub-section 2.6

The logistic regression analysis needs more description - were any confounding factors taken into account - what were they?

REPLY:Thank you very much for your suggestion. We have made revisions as follows

2.1 The group selection and flowchart havd been added.

2.2 The questionnaire and the validation havd been described in detail.

2.3 Specific year was added: as of March 31 2024, all pregnant women have completed their delivery

2.4 The exact classification of the beverages have been provided.

Simplify the description of macrosomia and move the relevant content to the discussion. This change ensures that the methodology section remains focused on the study design and procedures, while the results section addresses the study outcomes.

2.5 This part has been refined.

2.6 Added content about confounding factors in logistic regression analysis.

  1. Results.

Data visualisation: Use more visual and less text-heavy graphs and tables to make the results clearer and easier to understand.

 In Table 1, add N in the first row - header; are the data needed for x2 - consider whether it is better to delete these values.

Explanations:

*low-frequency: Greater than 0 time/month and less than or equal to 3times/week, high-frequency: greater than or equal to 4 times/week - should also be explained in the methodology.

The statement "The consumption rates of TB, SSB, NSS were as follows: 73.2%, 72.8% and 13.5%". - is too general or is it combined data for each consumption - the frequency of consumption should be analysed, especially daily.

REPLY:Thank you very much for your suggestion. The relevant content has been revised in the text. Considering that many studies have provided the x2, we hope that it can be retained.

  1. Discussion

I suggest that the manuscript be strengthened by paying more attention to the strengths and limitations of the study.

In addition, reference citations should be corrected according to the Guidelines for Authors. Please pay attention to item number 3 - a combination of 2 references.

Summary of Key Findings: Include a visual summary or infographic highlighting the study’s key findings to facilitate quick understanding of the most important points.

REPLY:Thank you very much for your suggestion. The relevant content has been revised in discussion. We also reviewed the references

Reviewer 2 Report

Comments and Suggestions for Authors

The authors describe their work on examining the health impacts of sugar-sweetened beverages (SSB) among pregnant women and their offspring. It was found that the consumption rates of total beverages (TB), SSB, and non-sugar sweetened beverages (NSS) were 73.2%, 72.8%, and 13.5%, respectively. Pregnant women consuming TB three times or less per week had a 38.4% increased risk of gestational diabetes mellitus (GDM) and a 64.2% increased risk of gestational hypertension (GH). Those consuming TB four or more times per week faced a 154.3% higher risk of GDM and 69.3% increased risk of GH. Regarding offspring health compared to non-consumers, TB consumption four or more times per week was associated with a substantial increase in the risk of macrosomia and large for gestational age (LGA). In the analysis of NSS, with a significantly increased risk of macrosomia, and LGA. The authors concluded that the high level of beverage consumption among pregnant women in Shanghai needs attention, and that excessive consumption of beverages increases the risk of GDM and GH, while excessive consumption of NSS possibly has a greater impact on offspring macrosomia and LGA. This is an interesting study. Appropriate methodology has been employed and the conclusions appear to be justified based on the data at hand. The authors are to be commended on the wealth of data presented. I have a few recommendations for consideration.

1.     Abstract. It would be helpful if the authors can state, in a sentence, the objective of their study.

2.     Introduction. Can the authors provide a clear hypothesis to be tested in the study?

3.     Methods/Results. Sugar is a general term, can the authors be more specific with respect to the composition of the SSB? i.e. fructose content?

4.     Results. Is 4,000 grams considered as the International standard for macrosomia?

5.     Results. The presentation of some of the data in the tables is difficult to read, can the authors improve?

6.     Results/Discussion. Aside from the birthweight, are there any other changes i.e. metabolic changes in the offspring such as insulin resistance, particularly in the offspring of mothers consuming high SSB and having GDM?

7.     Results/Discussion. How does the excessive consumption of NSS increase the risk for offspring macrosomia and LGA?

8.     Discussion. The authors need to emphasize and elaborate on the novelty aspect of their work as well as expand on the clinical applicability of their findings.

9.     Discussion. It would be helpful if the authors can describe some of the mechanisms that may be involved in the adverse maternal and offspring outcomes subsequent to SSB and NSS.

Author Response

  1. It would be helpful if the authors can state, in a sentence, the objective of their study.

REPLY:Thank you very much for your suggestion. The relevant content has been revised in abstract.

  1. Can the authors provide a clear hypothesis to be tested in the study?

REPLY:Thank you very much for your suggestion. The relevant content has been added.

  1. Methods/Results. Sugar is a general term, can the authors be more specific with respect to the composition of the SSB? i.e. fructose content?

REPLY:Related content added in the introduction.

  1. Is 4,000 grams considered as the International standard for macrosomia?

REPLY: Thank you very much for your suggestion. Macrosomia refers to growth beyond a specific threshold, regardless of gestational age. In high income countries, the most commonly used threshold is weight above 4500 g (9 lb 15 oz), but weight above 4000 g (8 lb 13 oz) is also commonly used. A grading system has been suggested grade 1 for infants 4000 to 4499 g, grade 2 for4500 to 4999 q, and grade 3 for over 5000 g.

  1. The presentation of some of the data in the tables is difficult to read, can the authors improve?

REPLY:Thank you very much for your suggestion. We have made some changes. If you think it is not suitable or have more suggestions, we will be happy to make more changes.

  1. Results/Discussion. Aside from the birthweight, are there any other changes i.e. metabolic changes in the offspring such as insulin resistance, particularly in the offspring of mothers consuming high SSB and having GDM?

REPLY:Thank you very much for your suggestion.We acknowledge the importance of investigating additional metabolic changes in the offspring. However, our study primarily focused on birth weight and the occurrence of macrosomia and large for gestational age (LGA). At this time, we do not have data on specific metabolic changes such as insulin resistance in the offspring. Future research will aim to address this limitation by including metabolic assessments in the offspring of mothers with high SSB consumption and gestational diabetes mellitus (GDM).

  1. Results/Discussion. How does the excessive consumption of NSS increase the risk for offspring macrosomia and LGA?

REPLY:Thank you very much for your suggestion. The possible mechanisms were explored in more detail.

  1. The authors need to emphasize and elaborate on the novelty aspect of their work as well as expand on the clinical applicability of their findings.

REPLY:Thank you very much for your suggestion. Related content has been refined.

  1. It would be helpful if the authors can describe some of the mechanisms that may be involved in the adverse maternal and offspring outcomes subsequent to SSB and NSS.

REPLY:Thank you very much for your suggestion. The possible mechanisms were explored in more detail.

Round 2

Reviewer 1 Report

Comments and Suggestions for Authors

The manuscript has been significantly improved.
However, the tables still lack readability. There are also no explanations below all the tables and Figure 2.
Maybe it will be done in the last revision.

Until the editor decides.

Best reagards.

Author Response

Comments:However, the tables still lack readability. There are also no explanations below all the tables and Figure 2.

Response: Thank you very much for your suggestion. I’m very sorry that the last revision was not perfect. We have deleted the x2 and made some other changes for better display the tables and Figure this time. If you think it is not suitable or have more suggestions or provide references, we will be happy to make more changes.